# Diversity of *Salmonella enterica* isolates from urban river and sewage water in Blantyre, Malawi

Jonathan Rigby[1,2,3¶]*, Catherine N. Wilson[2,4,5,6¶], Allan Zuza[2], Yohane Diness[2], Charity Mkwanda[2], Katalina Tonthola[2], Oscar Kanjerwa[2], Chifundo Salifu[2], Oliver Pearse[1,2], Chisomo Msefula[7], Blanca M. Perez-Sepulveda[5], Jay C.D. Hinton[5], Satheesh Nair[8], Nicola Elviss[8], Mathew A. Beale[6], Patrick Musicha[2,6,9], Nicholas A. Feasey[1,2,10]

1 Department of Clinical Science, Liverpool School of Tropical Medicine, Liverpool, United Kingdom, 2 Malawi-Liverpool-Wellcome Programme, Kamuzu University of Health Sciences, Blantyre, Malawi, 3 School of Public Health, Imperial College London, London, United Kingdom, 4 Queen's Veterinary School Hospital, University of Cambridge, Cambridge, United Kingdom, 5 Institute of Infection, Veterinary and Ecological Sciences University of Liverpool, Liverpool, United Kingdom, 6 Parasites and Microbes Programme, Wellcome Sanger Institute, Hinxton, Cambridgeshire, United Kingdom, 7 Kamuzu University of Health Sciences, Blantyre, Malawi, 8 United Kingdom Health Security Agency, Colindale, London, United Kingdom, 9 Vector Biology Department, Liverpool School of Tropical Medicine, Liverpool, United Kingdom, 10 School of Medicine, University of St. Andrews, St Andrews, United Kingdom

¶ These authors contributed equally
* Jrigby@ic.ac.uk

## Abstract

### Background

*Salmonella enterica* encompasses over 2,600 serovars, including several commonly associated with severe infection in humans. *Salmonella* is a major cause of sepsis in Africa; however, diagnosis requires clinical microbiology facilities. Environmental surveillance has the potential to play a role in *Salmonella* surveillance.

### Methods

We undertook water-based environmental surveillance in Blantyre, Malawi, from 2018-2020, taking samples from rivers (87.9%), a sewage plant (8.85%) and other water sources (3.24%), isolating and storing 1,042 non-typhoidal *Salmonella* (NTS) isolates in this period. Of these, 341 NTS isolates were whole genome sequenced, genome quality was checked, duplicate genomes from any given sample were removed and core genome phylogeny was reconstructed. AMRFinder, Pathogen-Watch and SISTR were used to further investigate serovar, sequence type and anti-microbial resistance determinants.

### Results

After quality checks, and removal of duplicate genomes, 270 NTS genomes remained for further analysis. Multiple *Salmonella* serovars associated with human

Data availability statement: European Nucleotide Archive: Genomes sequenced as part of the Drivers of antimicrobial resistance in Uganda and Malawi (DRUM) Project. Accession number PRJEB37378/ERP120687; https://www.ebi.ac.uk/ena/browser/view/PRJEB37378 BioStudies [Sarkans U, Gostev M, Athar A, Behrangi E, Melnichuk O, Ali A, et al. The BioStudies database-one stop shop for all data supporting a life sciences study. Nucleic Acids Res. 2018;46(D1):D1266-D70. doi: 10.1093/nar/gkx965]: Diversity of Salmonella enterica isolates from urban river and sewage water in Blantyre, Malawi. https://doi.org/10.6019/S-BSST1695. The above ENA Accession number and BioStudies DOI contains all of the relevant raw data discussed in this study, i.e. the Accession numbers of each isolate sequences and their result interpretation: Species, Serovar, Sequence Type and Antimicrobial profiles.

Funding: This work was primarily supported by the Bill and Melinda Gates Foundation through the Grand Challenges in Global Health initiative, awarded to Professor Nicholas Feasey (grant number INV-008749, https://www.grandchallenges.org/). This research was also supported in part by the Malawi-Liverpool-Wellcome Trust Programme, the institute where this study was conducted, which is funded under the Wellcome Asia and Africa Programme Grant 206545/Z/17/Z. The funders had no role in study design, data collection and analysis, decision to publish, or preparation of the manuscript.

Competing interests: The authors have declared that no competing interests exist.

infection were detected, of which *S.* Typhimurium (55/270 isolates) was the most common, including 44 of Sequence Type (ST) 313, a serovar commonly associated with severe invasive disease (iNTS). Six lineage 2 ST313 genomes possessed AMR genes predicting multidrug resistance (MDR), while 29 lineage 3 isolates contained no AMR predictive genes. PCR based detection of *staG* has been proposed as a diagnostic marker of *S.* Typhi; however, all eight genomes that contained *staG* identified as *Salmonella enterica* serovar Orion, raising concerns about the specificity of this marker as a monoplex for environmental surveillance of *S.* Typhi.

## Discussion

The study identified diverse *Salmonella* serovars in the environment, including those reported to cause invasive disease, emphasizing the complex but potentially valuable contribution of implementing environmental surveillance for *Salmonella* in high burden areas lacking diagnostic microbiology capacity.

## Author summary

*Salmonella enterica* is a diverse, complex species, with serovars associated with human, animal and environmental health, including invasive disease. This study sequenced isolates of non-typhoidal *Salmonella* (NTS) that had been cultured from river water and sewage during a typhoid environmental surveillance study in Blantyre, Malawi, between 2018–2020. We identified 43 different serovars, seven of which had two distinct sequence types. Three different subspecies: *S. enterica*, *S. salamae* and *S. diarizonae* were also identified. *S.* Typhimurium ST313 was the most prevalent, a sequence type (ST) commonly associated with invasive non-typhoidal *Salmonella* (iNTS) disease, with isolates from both lineage 2.0, a multi-drug-resistant lineage, and lineage 3, a drug-susceptible lineage also involved in many iNTS cases in Malawi. Other serovars of importance included extensively resistant *S.* Isangi, *fosA* containing *S.* Heidelburg and *staG* containing *S.* Orion. The prevalence of *staG* positive NTS serovars in the environment pose a challenge to monoplex PCR based surveillance using this gene target alone. This research emphasizes the value of environmental surveillance for *Salmonella* serovars in regions with limited diagnostic capabilities, where both typhoid fever and iNTS disease are of significant public health concern.

## Introduction

*Salmonella enterica* is a complex bacterial species with over 2,600 serovariants, or serovars [1]. *Salmonella* serovars have been implicated in both human and animal disease and can be responsible for a range of clinical pictures from asymptomatic colonisation, self-limiting enterocolitis, and life-threatening invasive disease [2]. Clinical presentation depends both on the properties of the serovar and the immune

status of the host. Salmonellae can form biofilms both within the body during infection and on environmental surfaces and they can colonise agricultural plants during growth [3,4].

*Salmonella* serovars are often loosely referred to by the human disease type they are most closely associated with, either as 'typhoidal' (*Salmonella enterica* serovar Typhi and Paratyphi A, B and C) or 'non-typhoidal' salmonellae (NTS). The prevalence of NTS as a cause of bloodstream infection (BSI) in Africa has led to the description of a distinct clinical syndrome called invasive NTS (iNTS) disease. Some serovars, including *S.* Typhimurium and *S.* Enteritidis, are more commonly associated with iNTS than other NTS serovars, due to specific lineages of ST313 *S.* Typhimurium and of ST11 *S.* Enteritidis [5–7].

The precise epidemiological burden of human disease caused by *Salmonella* in Africa has been difficult to ascertain as diagnosis depends on the availability of quality assured diagnostic microbiology capacity [8]. Where surveillance has been established, both typhoidal *Salmonella* (inclusive of *S.* Typhi and *S.* Paratyphi A, B and C) and NTS serovars have often been identified as common causes of BSI [9]. This has led to efforts to prioritise *Salmonella* vaccine development, including the recent Typhoid Conjugate Vaccine (TCV) [10]. For public health services to appropriately target the rollout of TCV, it is important to understand where *S.* Typhi is prevalent [11]. Environmental surveillance has been proposed as a tool to identify the presence of *S.* Typhi in settings where diagnostic blood culture capacity is limited [12–15]. Whilst blood culture remains the gold standard for diagnostics, a major development has been the introduction of real-time PCR (qPCR) for the identification of the serovar. One gene target frequently used is the *staG* gene [16]. Originally proposed as sensitive and specific for the identification of *S.* Typhi in clinical samples, subsequent work challenged this [16,17].

Detection of *S.* Typhi in the environment and discrimination from other serovars of *Salmonella* is challenging [18–20]. Between 2018 and 2020 we developed field and laboratory methods for the identification of salmonellae in the environment in Blantyre, Malawi. As a part of this programme, we ran a pilot study in areas of the city with reported high incidences of typhoid BSI cases confirmed by blood culture [21,22]. We demonstrated that although challenging, culture of *S.* Typhi from natural river water is possible, [18]. The identity of single *Salmonella* colonies were confirmed by qPCR [17,18], API20E and *Salmonella* anti-sera according to the Kaufmann and White Scheme [23, 24]. Because this work relied on *Salmonella enterica* selective media (mCASE, NCM1016A, Neogen), significant numbers of NTS were contemporaneously isolated.

The aim of this research was to use whole genome sequencing to determine the diversity of NTS isolated from water systems in Malawi with three specific objectives; firstly, to identify key human pathogenic serovars and multi-locus sequence types circulating in the environment, secondly, to establish which antimicrobial resistance determinants are present in salmonellae circulating in the environment. Thirdly, having isolated phenotypically confirmed NTS that gave positive amplification for gene targets previously proposed as being specific for *S.* Typhi, we aimed to identify which serovars were responsible for qPCR cross-reactivity.

## Methods

### Ethics statement

This study was completed under ethics application P.10/19/2819, ethical waiver P.07/20/3089 from the University of Malawi College of Medicine Research Ethics Committee (COMREC), now part of Kamuzu University of Health Sciences.

### Study design

From here onwards, two terms are used which are defined as:

- **Samples**: The specific, whole water specimen that was collected from study sites.

- **Isolate**: During culture, up to 10 colonies could be selected for purification. Isolates are the cultures of these colonies that conformed to *Salmonella* spp. morphology.

The environmental surveillance study team collected 2,693 samples of water, Moore swabs, soil and biofilms over an nineteen-month period between June 2018 to January 2020. Sites were selected initially based on water usage points close to predicted typhoid clusters based on geolocation data from patients admitted to Queen Elizabeth Central Hospital with culture confirmed typhoid fever using the analysis from Gauld *et al.,* 2022 [21,22] plus the inclusion of a wastewater treatment plant. A map of Blantyre with the sampling locations can be found in Fig 1

The method for sample collection and processing used is described in detail as the "Pathway P" approach published in Rigby *et al.*, 2022 [18] and a version can be found on Protocols.io [25]. In brief, samples collected were categorised as either composite samples, exposed to the water source over a longer period of time (Moore's Swabs [26], soil/sediment samples and biofilms) and grab samples, snapshots of the water flowing through at the time of collection (one-litre water samples were collected from rivers, a wastewater plant, boreholes, water kiosks, and market produce washing buckets).

Water samples were filtered through a 0.45 µM cellulose nitrate membrane (Ref. 515–0228, Sartorius). Filter papers and the other sample types were all incubated in bile⁻ broth (Ingredients per one litre of distilled water: 20g Ox Bile [Ref. NCM0240A, Neogen]; 5g Dextrose [Ref. NCM0241A, Neogen]; 10g Peptone [Ref. 70176, Merck]; 8g Sodium phosphate [Ref. 71643, Merck]; 2g Potassium dihydrogen phosphate [Ref. NIST200B, Merck]). Samples were incubated at $37 \pm 1$ °C for $18 \pm 2$ h. Post-incubation, 5 mL of bile⁻ broth was transferred to double strength selenite F broth (38g Selenite Broth Base [Ref. CM0395, Oxoid, Basingstoke, UK] and 8g Sodium Biselenite [Ref. LP0121, Oxoid] [27]) in glass tubes and incubated at $41 \pm 1$ °C for $18 \pm 1$ h. Samples were plated on mCASE, diluted in Ringer's Lactate Solution (Ref. BR0052, Oxoid), and incubated. Blue/green colonies were confirmed by qPCR.

DNA was extracted using the thermal lysis ("boilate" method) in UltraPure DNase/RNase-Free Distilled Water (Ref. 10977035, Thermofisher Scientific Invitrogen). Bacterial colonies were suspended in nuclease-free water, heated at 96°C for 10 minutes, and pulse centrifuged (16,000 g for 5 seconds) to lyse cells.

Isolates were screened using a qPCR adapted from Nair *et al.*, 2019 [17], adopting only the gene targets relevant for *S.* Typhi detection (S4 Table [18,19,28]). This assay used a triplex assay targeting *ttr*, a pan-*Salmonella* gene, found in all subspecies and serovars of both species of *Salmonella*: *enterica* and *bongori*, involved in tetrathionate respiration (AF282268 [29]), *tviB*, a *S.* Typhi specific component gene of the *viaB* locus, which is involved in the synthesis, transport and expression of the Vi antigen (NC_003198 [17]) and *staG*, also an *S.* Typhi specific gene designed for use with blood culture for diagnostics, which is involved in fimbriae production (AL513382 [16]).

Isolates which were *ttr* positive were archived as NTS, whilst any isolates that contained *staG* or *tviB* were then further verified using a biochemistry test for Enterobacteriaceae, that could distinguish between NTS and typhoidal *Salmonella* spp. (BioMérieux API 20E), and serology using the Kaufmann-White scheme, (O12, O9, Vi and Hd) to confirm whether they were *S.* Typhi, *S.* Typhimurium, *S.* Enteritis or other NTS [24,30–32].

A total of 1,048 *Salmonella enterica* isolates were obtained from 425 culture positive samples, including six *S.* Typhi, identified by the qPCR, biochemistry and serology detailed above. The remaining 1,042 isolates were confirmed to be NTS using previously published qPCR primers; specifically isolates that were qPCR positive for the pan-*Salmonella* marker *ttr* alone or with either (but not both) *tviB* and *staG* additionally being qPCR positive (Table 1).

Three-hundred-forty-one isolates were selected for whole genome sequencing (WGS). From the six samples that yielded isolates with positive amplification for all three primer targets we selected all 19 discreet single colonies. As the focus of this study is NTS, whilst these isolates had the gene targets for *S.* Typhi, this data set will only discuss those isolates that yield NTS from sequencing. Additional isolates of interest were chosen based on qPCR results, including all NTS isolates that were *ttr* positive plus either *staG* (n = 23) or *tviB* (n = 1) positive. A further 296 isolates were selected at random, attempting to ensure only one colony pick from each environmental sample chosen was submitted for WGS.

Of the 341 isolates selected for sequencing, 187 were grab samples, from rivers (n = 154) and wastewater (n = 33). Moore's swabs accounted for 108 samples from rivers (n = 66) and wastewater (n = 42). The remainder 46 isolates were from biofilms and algae (n = 24), soil and sediment samples (n = 18) and food (n = 4).

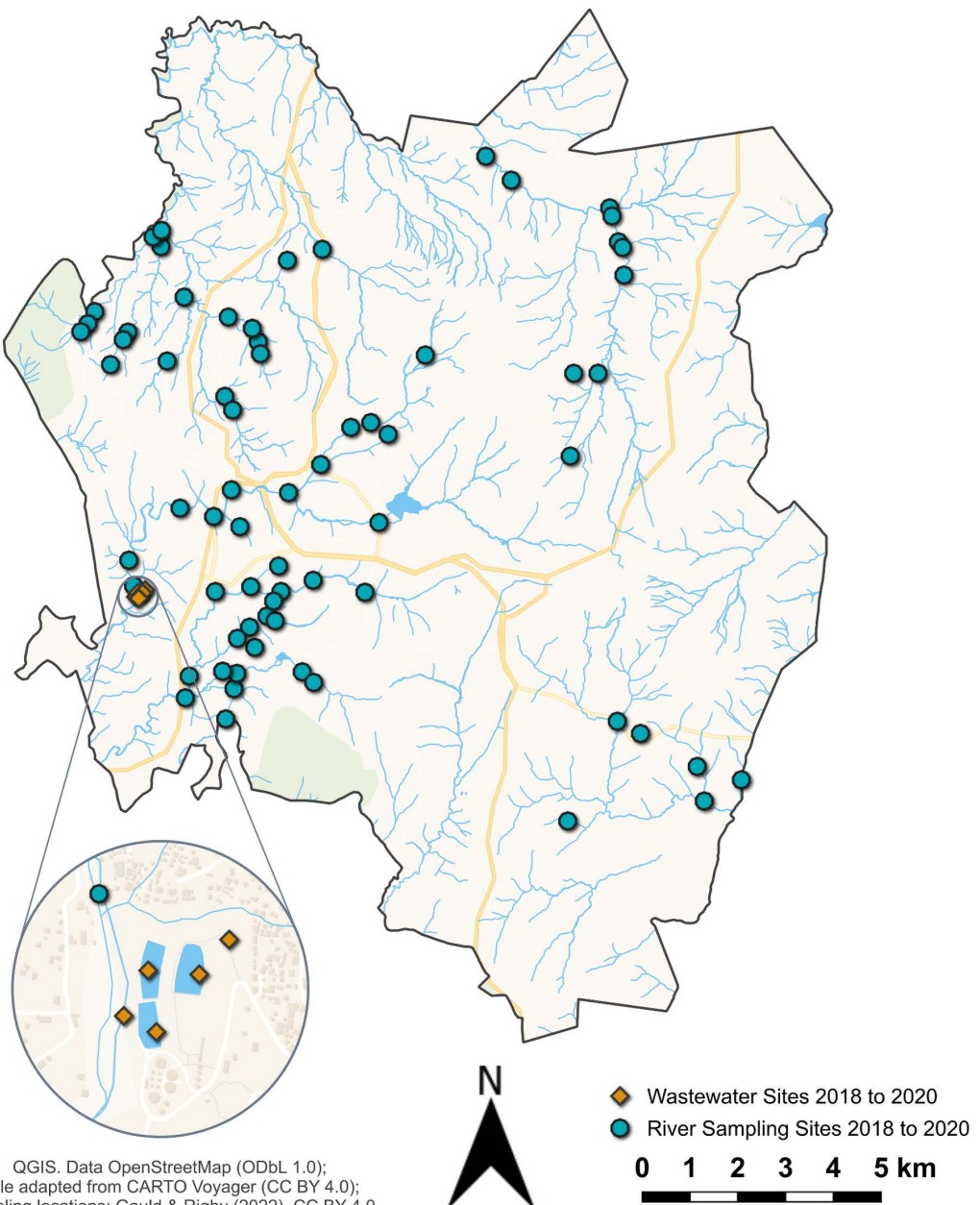

## Locations of sampling sites used in Blantyre, Malawi, 2018 to 2020

◆ Wastewater Sites 2018 to 2020

● River Sampling Sites 2018 to 2020

0  1  2  3  4  5 km

QGIS. Data OpenStreetMap (ODbL 1.0);
Style adapted from CARTO Voyager (CC BY 4.0);
Sampling locations: Gauld & Rigby (2022), CC BY 4.0.

**Fig 1. Locations of all sampling sites used in Blantyre, Malawi, during water surveillance between 2018 and 2020, depicting both natural water (•) and wastewater (♦) sites.** Map created using the free and open-source software QGIS (https://qgis.org). Tiles were generated using Tilemaker (https://tilemaker.org) from OpenStreetMap data by OpenStreetMap contributors (https://www.openstreetmap.org/copyright), licensed under the Open Database License (ODbL 1.0) (https://opendatacommons.org/licenses/odbl/1-0/). Map style adapted from the Voyager stylesheet by CARTO (https://github.com/CartoDB/basemap-styles), licensed under the Creative Commons Attribution 4.0 International license (https://creativecommons.org/licenses/by/4.0/). District boundaries derived from the GADM database of Global Administrative Areas (https://gadm.org/license.html), used with permission for academic publication. Sampling locations are described in Gauld et al., (2022) [21,22] and Rigby et al., (2022) [18].

**Table 1. Distribution of *Salmonella enterica* isolated from the environment of Malawi 2018-2020 selected for whole genome sequencing based on Real-Time Polymerase Chain Reaction (qPCR) based serovar prediction.**

| qPCR results combination* | Number of genome-sequenced isolates (Number of Samples from which at least one isolate was sequenced) | River Water - Grab | River Water - Moore's Swab | Wastewater - Grab | Wastewater - Moore's Swab | Soil/ Sediment | Biofilm/ Algae | Food |
|---|---|---|---|---|---|---|---|---|
| NTS‡ (*ttr*-positive, *tviB* and *staG*-negative) | 298 (277) | 132 | 57 | 33 | 33 | 18 | 21 | 4 |
| NTS‡ (*ttr* and *staG*-positive, *tviB*-negative) | 23 (8) | 20 | 1 | – | 1 | – | 1 | – |
| NTS‡ (*ttr* and *tviB*-positive, *staG*-negative) | 1 (1) | – | – | – | – | – | 1 | – |
| Suspected *S.* Typhi (*ttr*, *tviB* and *staG*-positive) | 19 (6) | 2 | 8 | – | 8 | – | 1 | – |
| Total | 341 (292) | 154 | 66 | 33 | 42 | 18 | 24 | 4 |

*Primer combinations targeting genes *ttr*, *tviB*, and *staG* were used to infer potential serovar identity, where *ttr*, *tviB*, and *staG* indicated presumptive *S.* Typhi isolates, *ttr* positivity plus either *tviB* or *staG* may have been *S.* Typhi and *ttr* alone were likely other Salmonella serovars.

‡Non-Typhoidal *Salmonella*

Isolates were stored at -80°C in an ultra-low temperature freezer (ULT) on latex beads with a glycerol buffer (PL.170C, Microbank). The isolates for whole genome sequencing were recovered from the freezer by inoculation of a bead onto mCASE media, streaking for single colonies. Single colonies from archived samples were picked and sub-cultured onto nutrient agar (CM0003, Oxoid) to ensure both purity and that only a single strain was sent for whole genome sequencing. Purification was necessary due to the beads often containing multiple salmonellae with different serological profiles when routine screening for *S.* Typhi was performed.

## Sample preparation and extraction of genomic DNA

A pre-lysis step was performed by taking a 1 µL loop of bacterial growth from nutrient agar and inoculating 1.5 mL of nutrient broth, which was incubated at $37 \pm 1°C$ for 18–20 hours. After incubation, samples were heat inactivated at $95 \pm 2°C$ for 10 minutes, with 700 µL transferred into a deep 96-well plate (4titude). The plates were centrifuged for 20 minutes at 2,500 x g. The supernatant was discarded and replaced with 220 µL of ATL cell lysis buffer (939016, Qiagen) and 20 µL proteinase K (19133, Qiagen), and incubated at $60 \pm 5 °C$ for 30 minutes, 4 µL of RNase was added and incubated at $37 \pm 1°C$ for 15 minutes. Lastly, as per manufacturer's instructions, the plate was loaded onto the QiaSymphony (9001301, Qiagen), and a DSP Virus/Pathogen mini-Kit (937036, Qiagen) was used with the default extraction profile on the machine. Yield and purity of each genomic DNA sample after extraction was determined using the Qubit (Q33238, Thermofisher Scientific) 1x dsDNA Broad Range Assay kit (Q33230, Thermofisher Scientific).

## Whole genome sequencing and quality control

The genomic DNA was sent to the Wellcome Sanger Institute (WSI) under the terms of a Nagoya Protocol-compliant Access and Benefit Sharing Agreement (ABS1631659402922). At WSI, library preparation was performed using NEB Ultra II custom kit on an Agilent Bravo WS automation system. Whole genome sequencing was performed on the Illumina HiSeq X10 platform (Illumina Inc, California, USA) to generate paired-end raw reads of 150 base pairs (bp). FastQC (https://www.bioinformatics.babraham.ac.uk/projects/fastqc/, version 0.11.9) and multiQC (https://multiqc.info/, version 0.11.8) were used to assess quality of raw reads [33]. We performed species confirmation with Kraken (version 1.1.1), excluding genomes <70% abundance of *Salmonella* reads [34]. Sequences from this study, along with reads published elsewhere and used for context, were trimmed using Trimmomatic [35] (version 0.39).

Raw reads were assembled into contiguous sequences (contigs) and annotated using SPAdes v3.15.5 [36] and PROKKA v1.14.5 [37], respectively, via a WSI automated pipeline [38]. Quality of the genome assemblies was assessed using first CheckM v1.1.2 [39], to assess contamination and completeness of the genomes, using an exclusion threshold of >20% contamination and <90% completeness (version 1.1.2) [39]. The Quality assessment tool for genome assemblies (QUAST) was used to assess the number of contigs, N50 and total length of the genome, using an exclusion threshold of contigs >500, N50 < 20kb, and total base pairs <4Mbp or >5.8Mbp (version 5.0.2) [34,40]. Following the completion of quality control procedures, genomes were submitted to PathogenWatch [41] (https://pathogen.watch/), which uses SISTR (sistr_cmd, version 1.1.1 [42,43]) to assess the species, serovar and sequence type of the bacteria present. All tools were used with the standard, default parameters. Genomes were released to the European Nucleotide Archive per WSI Open Access Policy, under project ID PRJEB37378.

All assemblies were screened directly for predicted amplicons within the expected size range between the primer sequences in_silico_pcr (available at https://github.com/sanger-pathogens/sh16_scripts/blob/master/legacy/in_silico_pcr.py) was used.

## Core-genome phylogeny and single nucleotide polymorphism analysis

A core and pangenome analysis were performed using Roary [44], (version 3.11.2). A core gene sequence alignment was generated by concatenating the alignments of the core genes. A gene was considered core if it was present in 100% of the genomes at a match identity threshold of 95% (241). A single nucleotide polymorphic (SNP) site alignment was extracted from the core gene alignment using SNP-sites [45], (version 2.5.1). RAxML (Randomised Accelerated Maximum Likelihood) (version 8.2.8 [46]) was run on the resulting core SNP-alignment to construct a maximum likelihood tree using the core gene SNP alignment of all isolates passing QA. Reliability of inferred branch partitions was assessed with 100 bootstrap replicates using the Infinitely Many Genes model as part of Panaroo [47,48]. The tree was visualised using ggtree [49] (version 3.2).

During isolation, any sample that yielded multiple colonies with *Salmonella* spp. morphology had up to 10 colony "picks" sub-cultured due to the low abundance of *S.* Typhi within positive samples, and number of competing salmonellae in each sample, therefore, it was assumed that if the two isolates were genomically identical and from the same sample, then they were subcultures of the same organism. A 'duplicate' was defined as two genomes, collected from the same sample, which were between 0–2 pairwise SNP distance within the core gene alignment calculated using snp-dists (https://github.com/tseemann/snp-dists) [50], and therefore consequently also the same serovar and ST type. 'Duplicated' genomes were removed from the tree. The tool snp-dists [51] was run on the core gene alignment (version 0.7.0). One isolate of each duplicate genome pair collected from the same water sample was removed from the RAxML tree using the `drop.tip` command in the ape package in R [52,53]. Genomes with a pairwise SNP distance of 3 SNPs or greater were maintained in the tree.

## Reference mapping S. Typhimurium ST313 and S. Enteritidis ST11 to contextual genomes

A reference mapping approach was used to relate to the study isolates to contextual isolates using the Burrow-Wheeler Alignment (BWA) tool (version 0.7.17). The reference genomes for *S.* Typhimurium ST313 was D23580 (Accession Number GCA 009953275 FN424405 [54]). The reference genome *S.* Enteritidis was P125109 (Accession Number GCA 000009505 AM933172 [55]). The sequences for these reference genomes were available in the WSI repositories. SNP-sites was used to identify nucleotide difference between isolates ( [45], version 2.5.1). A multi-sequence alignment of reference-based pseudo-genomes was used to infer a maximum likelihood phylogeny using RAxML (version 8.2.8) with 100 bootstrap replicates to assess support. Phylogenetic trees were rooted with a suitable serovar or sequence type; either a previously published reference genome for that serovar, or a distantly related *Salmonella enterica* isolate genome. Visualisations were performed with ITOL [56], (version 6.5.7).

The lineage of *S.* Typhimurium ST313 and *S.* Enteritidis ST11 isolates were classified on the basis of their relationship with a selection of published contextual genomes (Available in S5 and S6 Tables) within these phylogenetic trees, presented in S1 and S2 Figs. A comparison of the relation of pairwise SNP distance measurements of closely related genomes (within the same lineage) in regard to their position within the phylogenetic tree aided assessment that the correlation with the contextual lineage was correct.

### Identification of antimicrobial resistance (AMR) determinants, virulence factors, and plasmid typing

AMRFinderPlus (version 3.1010) was used to detect chromosomal mutations encoding for AMR, acquired AMR genes (ARGs), and heavy metal resistance genes [57]. Those ARGs with an identity of >95% and a coverage of >95% were taken forward for further analysis.

## Results

In total 301/341 (88.2%) genomes passed quality checks. Duplicate genomes (30/301) that originated from the same water sample were removed from the core gene alignment. In addition, one additional *S.* Typhi genome was identified and removed from the alignment, leaving a total of 270 NTS genomes for phylogenetic reconstruction. Fifteen discrete samples contained a mixture of genomes that originated from more than one distinct serovar, multi-locus sequence type (MLST) or were more than two SNPs different within the core gene alignment (S2 Table).

The collection of 270 genomes formed three phylogenetic clusters (Fig 2), corresponding to three of the five subspecies of *Salmonella enterica*. No isolates of *S. enterica* subsp. *arizonae*, *houtenae* or *indica* were identified. Neither were there any isolates of the other salmonellae species, *S. bongori*.

The majority of the genomes (261/270, 97%) were *Salmonella enterica* subspecies *enterica (S. enterica);* 6/270 (2%) were subspecies *salamae* (*S. salamae*) and 3/270 (1%) were subspecies *diarizonae* (*S. diarizonae*). A diverse array of 43 serovars were identified across the subspecies. Thirty-seven previously recorded serovars of *S. enterica* were described including *S.* Typhimurium, *S.* Enteritidis, *S.* Heidelberg, *S.* Oranienburg, *S.* Isangi and *S.* Amager (Table 2). In total, five different antigenic profiles of *S. salamae* were present, whilst the three *S. diarizonae* shared the same predicted antigenic profile.

Overall, the most common MLST was *S.* Typhimurium ST313, which represented 44/270 (16.3%) genomes and was detected in 13/19 (68.4%) months of the sampling period. Fourteen *S.* Typhimurium ST313 genomes were most closely related to lineage 2 (Africa and iNTS disease associated, genotypically multidrug resistant [MDR, as defined by resistance to three or more antimicrobial classes]), and 29 were most closely related to lineage 3 genomes (Africa and iNTS disease-associated, no predicted AMR genes) [58]. One of the ST313 genomes does not appear to be closely related to any previously documented clade (S1 Fig). The 44 *S.* Typhimurium ST313 (Table 2, Fig 2) isolates were compared to other contextual sequences (S5 Table and S1 Fig) including a representative selection of ten of each of lineage 1 (African invasive non-multidrug resistant isolates), lineage 2 (African invasive multidrug resistant isolates), lineage 2.1 (African invasive multidrug resistance sub-lineage found in DRC), lineage 3 (African (Malawian) pan-susceptible lineage) and UK-like isolates [6,58–61]. Seven *S.* Enteritidis ST11 genomes were predicted to be part of the "outlier cluster" of ST11, from which the Global Epidemic Clade emerged [5] (S2 Fig). Sixteen *S.* Isangi genomes were present in the collection, all of which showed not only a MDR genotype, but also contained ESBL determinants) (Figs 2 and 3, and Table 3); these are of interest due to a contemporaneous outbreak on the neonatal unit at Queen Elizabeth Central Hospital (QECH) Blantyre, Malawi (O. Pearse, personal communication).

Other identified serovars previously associated with iNTS disease included [62] *S.* Heidelberg (22/270, 8.1%) and as the next most frequently identified; *S.* Hadar (6/270, 2.2%), *S.* Bovismorbificans (3/270, 1.1%) and *S.* Infantis (1/270, 0.4%) and *S.* Rubislaw (1/270, 0.37%). There were also four isolates identified as being potentially *S.* Hissar, *S.* Choleraesuis, *S.* Paratyphi C, *S.* Typhisuis, *S.* Chiredzi or *S.* Rubislaw, three of which were ST3965, whilst the other could not be identified further using SISTR.

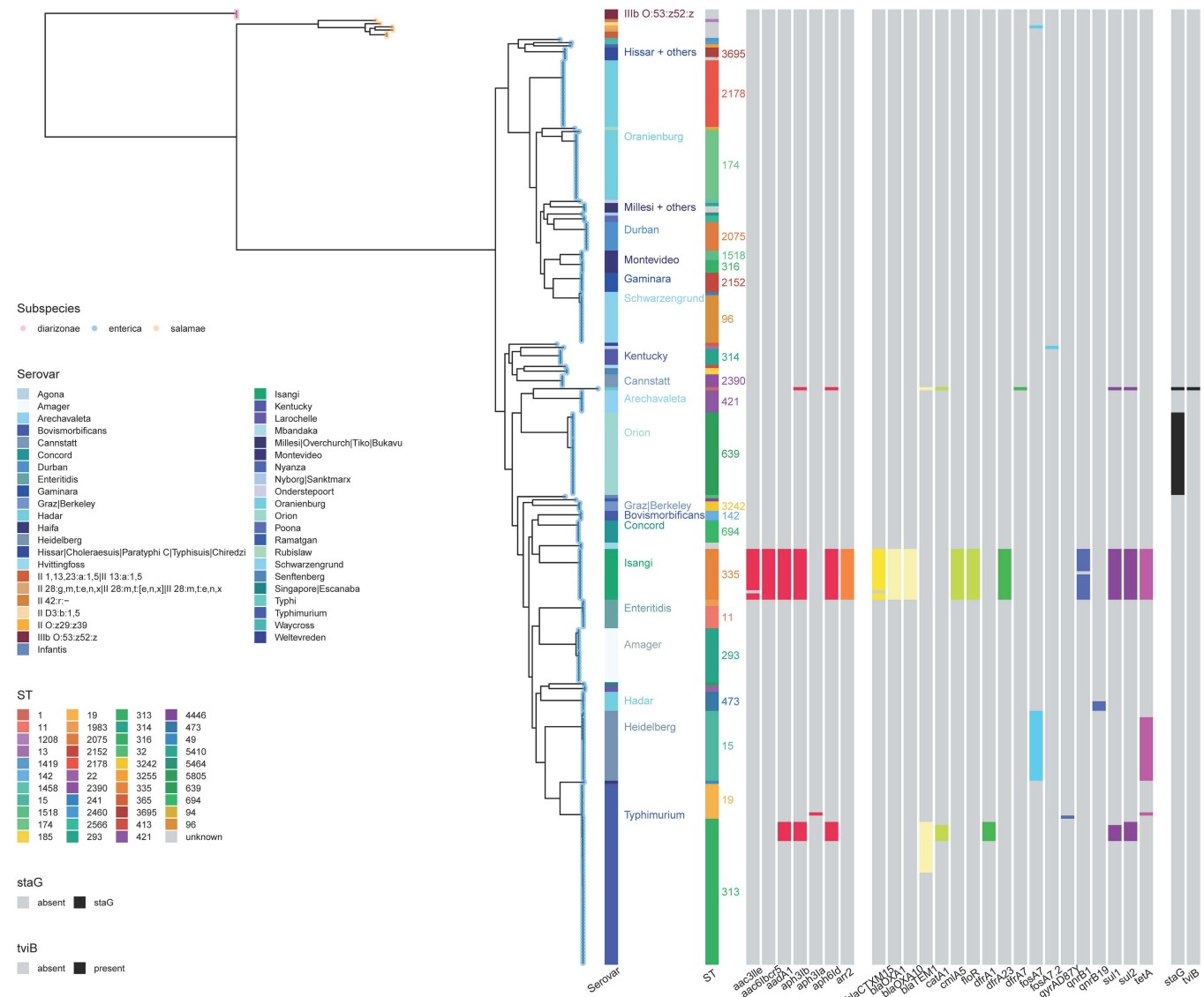

**Fig 2. Maximum likelihood RAxML phylogenetic tree demonstrating the relationship of *Salmonella* isolates within this study (n = 270).** The tree was constructed using a core gene alignment in RAxML, midpoint rooted and visualised using ggtree [46,49]. Coloured tree tip labels denote *Salmonella* subspecies. Coloured tracks indicate serovar, MLST and presence/absence of antibiotic resistance genes.

## Antimicrobial resistance patterns

Overall, 59/270 (21.9%) genomes had at least one genotypically predicted AMR mechanism (Fig 3, Table 3). Five genomes had the MDR pattern typically associated with human NTS isolates in Africa (resistance to ampicillin, chloramphenicol and co-trimoxazole) [63]. Seventeen genomes were MDR, of which 15 were *S*. Isangi ST335 which carried ESBL genes (*blaCTX-M-15*, *blaOXA-1* and *blaOXA-10*).

Since its first discovery, genomes of the sequence type ST313 has been further differentiated into lineages and sublineages. Chloramphenicol-sensitive ST313 lineage 1 was replaced in Malawi by chloramphenicol-resistant lineage 2 [59]. Subsequently, there has been emergence of ST313 sub-lineage 2.i within the Democratic Republic of Congo [6]. ST313

**Table 2. Frequency of the 270 *Salmonella enterica* sequenced, by serovar and sequence type (ST) isolated from water samples taken from the environment in Malawi (2019-2020), using the SISTR pipeline for serovar identification.**

| Serovar | ST | Number | Percentage |
|---|---|---|---|
| Typhimurium | 313 | 44 | 16.30% |
| Heidelberg | 15 | 22 | 8.15% |
| Oranienburg | 174 | 21 | 7.78% |
| Oranienburg | 2178 | 20 | 7.41% |
| Isangi | 335 | 16 | 5.93% |
| Amager | 293 | 14 | 5.19% |
| Typhimurium | 19 | 11 | 4.07% |
| Schwarzengrund | 96 | 11 | 4.07% |
| Orion | 639 | 9 | 3.33% |
| Durban | 2075 | 9 | 3.33% |
| Enteritidis | 11 | 7 | 2.59% |
| Concord | 694 | 7 | 2.59% |
| Arechavaleta | 421 | 6 | 2.22% |
| Hadar | 473 | 6 | 2.22% |
| Gaminara | 2152 | 6 | 2.22% |
| Kentucky | 314 | 5 | 1.85% |
| Montevideo | 316 | 4 | 1.48% |
| Cannstatt | 2390 | 4 | 1.48% |
| Bovismorbificans | 142 | 3 | 1.11% |
| Montevideo | 1518 | 3 | 1.11% |
| Graz \| Berkeley* | 3242 | 3 | 1.11% |
| Hissar \| Choleraesuis \| Paratyphi C \| Typhisuis \| Chiredzi \| Rubislaw* | 3695 | 3 | 1.11% |
| IIIb O:53:z52:z | unknown | 3 | 1.11% |

*These isolates could not be further differentiated to identify specific serovars. For clarity, this table was reduced by removing sequence types with 2 or less isolates. The full list can be found in S3 Table.

lineage 3, which is antibiotic-sensitive, emerged in 2016 [58]. Further phylogenetic analysis of lineage 2 isolates have documented the presence of the sublineages 2.2 and 2.3 which emerged between 2006–2008 and have been replacing lineage 2.0 [61].

The two AMR serovars/STs of note were *S*. Typhimurium ST313 (lineage 2) and *S*. Isangi. ST313 lineage 2 *S*. Typhimurium genomes were predictive of MDR for 6/14 isolates, whilst 8/14 only carried genes associated with resistance to aminopenicillins. ST313 lineages previously described as being in circulation in Blantyre, include lineages 2.0, 2.2, 2.3 and lineage 3. Whilst lineage 3 is typically antimicrobial susceptible, lineages 2.0, 2.1, 2.2 and 2.3 are MDR. Lineage 2.1 has previously only been described in the Democratic Republic of Congo [6], whilst 2.0 was identified in Blantyre previously [64], 2.2 and 2.3 were identified from a large collection of Blantyre clinical isolates [5], having emerged in Malawi in 2006 and 2008 respectively [61,65]. Twenty-nine ST313 lineage 3 genomes contained no obvious AMR determinants [61]. This lineage was determined by comparing the "unknown" sequence types to published contextual strains, with these 29 being closely related to those published in Pulford *et al*., 2021 (S1 Fig) [58].

None of the seven *S*. Enteritidis carried detectable AMR determinants. All 16 *S*. Isangi genomes contained multiple AMR gene variants; including at least one AMR determinant conferring resistance to aminoglycosides, rifampicin, aminopenicillins, chloramphenicol, trimethoprim, sulphonamide and tetracycline, fluoroquinolones and extended spectrum beta-lactamases (Fig 2, Table 3).

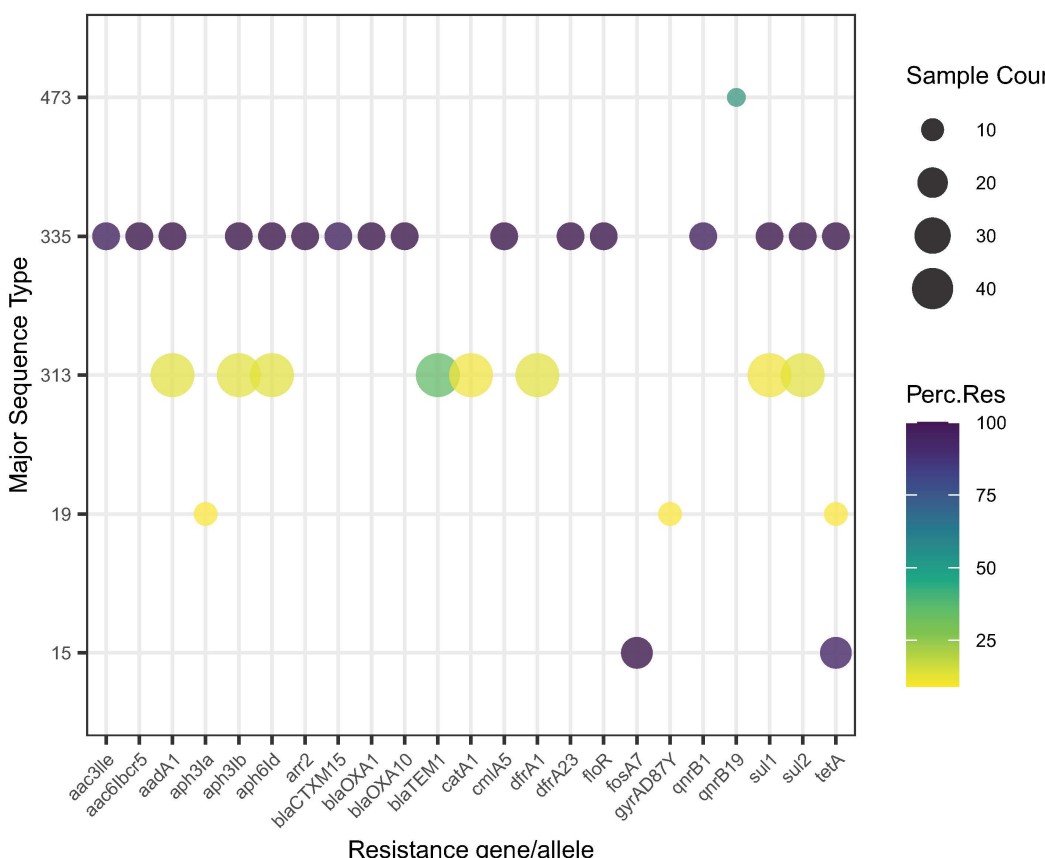

**Fig 3. Bubble plot demonstrating the number and proportion of *Salmonella* sequence types carrying specific genomic antimicrobial resistance determinants.** Within the collection (n = 270) antimicrobial resistance determinants are mainly carried by salmonellae of five sequence types, which are displayed within this figure (*S*. Typhimurium ST313, *S*. Isangi ST335, *S*. Hadar ST473, *S*. Typhimurium ST19, *S*. Heidelberg ST15). Sample count is denoted by the diameter of the bubble and percentage of isolates of each sequence type which carry the specific genomic resistance determinant is denoted by the colour of the bubble.

**Table 3. Frequency of predicted antimicrobial resistance (AMR) genes and markers within 103 *Salmonella enterica* genomes isolated from the environment in Malawi 2018-2020 containing at least one antimicrobial determinant.**

| Serovar (No. of isolates) | Amino-glycosides | Rif-ampicin | Extended-Spectrum Beta Lactamase | Amino-penicillins | Chloram-phenicol | Trimetho-prim | Fosfomy-cin | Fluro-quinolone | Sulphon-amides | Tetra-cyclines |
|---|---|---|---|---|---|---|---|---|---|---|
| *S*. Isangi (16) | 16 (100%) | 16 (100%) | 15 (93.75%) | 16 (100%)* | 16 (100%) | 16 (100%) | 0 (0%) | 15 (93.75%) | 16 (100%) | 16 (100%) |
| *S*. Typhimurium ST313 & ST19 (57) | 7 (12.28%) | 0 (0%) | 0 (0%) | 16 (28.01%) | 5 (8.77%) | 6 (10.53%) | 0 (0%) | 1 (1.75%) | 6 (10.53%) | 1 (1.75%) |
| *S*. Heidelberg (22) | 0 (0%) | 0 (0%) | 0 (0%) | 0 (0%) | 0 (0%) | 0 (0%) | 22 (100%) | 0 (0%) | 0 (0%) | 20 (100%) |
| *S*. Hadar (6) | 0 (0%) | 0 (0%) | 0 (0%) | 0 (0%) | 0 (0%) | 0 (0%) | 0 (0%) | 3 (50%) | 0 (0%) | 0 (0%) |
| *S*. Agona (1) | 0 (0%) | 0 (0%) | 0 (0%) | 0 (0%) | 0 (0%) | 0 (0%) | 1 (100%) | 0 (0%) | 0 (0%) | 0 (0%) |
| *S. salamae* II O:z29:z39 (1) | 0 (0%) | 0 (0%) | 0 (0%) | 0 (0%) | 0 (0%) | 0 (0%) | 1 (100%) | 0 (0%) | 0 (0%) | 0 (0%) |

*Based on resistance to aminopenicillins being conferred by *blaCTX-M-15*.

Whilst not all *S*. Heidelburg genomes were predicted to be MDR, all 22 sequenced genomes of this serovar carried the *fosA7* AMR gene which confers genotypic resistance to fosfomycin, as did *S*. Agona and *S*. *salamae* II O:z29:z39. The majority of the *S*. Heidelberg isolates (19/22) also carried the *tet(A)* tetracycline resistance gene. Half of the *S*. Hadar isolates (3/6) carried *qnrB19*, the AMR gene which confers quinolone resistance.

## Prevalence of staG positive NTS serovars in the environment

Sixteen of the 1,048 non-typhoidal *Salmonella* (1.4%) colonies isolated from 425/2,693 culture-positive samples were qPCR-positive for *staG.* Following qPCR, the 270 genomes were screened for the presence of *staG* and the only sero-var which was proved to contain *staG* was *S*. Orion (seven genomes, Table 4, Fig 2). The remaining eight qPCR positive isolates were positive for *staG* lacked the gene when sequenced, as per Table 4.

The nine *S*. Orion genomes originated from seven unique samples, two samples of which had two genomes of *S*. Orion that were more than three SNPs different to one another and were kept as separate genomes for analysis. From the six samples that had confirmed *S*. Typhi isolates, we additionally isolated and sequenced 13 NTS genomes; of those, four were *S*. Orion. Eight of the nine *S*. Orion isolates were positive by qPCR (*in vitro*) for *ttr* and *staG* and when tested by biochemistry and serology to determine whether they were *S*. Typhi during the environmental surveillance, they were determined to be NTS. The ninth was qPCR negative and so not tested further before sequencing; however, *in silico* PCR for *staG* and *tviB* performed on all genome assemblies confirmed that all nine *S*. Orion genomes contained *staG*, and that only *S*. Orion from our collection contained *staG*. Further isolates from different serovars were *staG* positive by qPCR *in vitro*, but unlike *S*. Orion, none of these contained *staG* in their genomes.

## Discussion

Analysis of 270 unique NTS genomes isolated from a 19-month period of environmental surveillance, identified a diverse collection of 43 serovars. Among the human disease-causing serovars, those of significant local public health concern included *S*. Typhimurium ST313 Lineage 2, *S*. Enteritidis and MDR and ESBL *S*. Isangi.

This is the first time that *S*. Typhimurium ST313 has been detected from environmental water sources in Malawi within river and sewage water around Blantyre. Finding *S*. Typhimurium ST313 in water sources raises the possibility that urban river systems may play a role facilitating transmission of this key, iNTS disease-associated sequence type to humans. ST313 is the most common sequence type of *S*. Typhimurium to be isolated from bloodstream infections in sub-Saharan Africa [7], however, its transmission routes remain unclear. It is generally assumed that it has a human reservoir and that person to person transmission dominates, however these findings raise the possibility that ST313 has a long-cycle trans-mission cycle akin to that of *S*. Typhi [66,67].

**Table 4. Distribution of Real-Time Polymerase Chain Reaction (qPCR) targets thought to be specifically associated with *Salmonella* Typhi amongst non-typhoidal *Salmonella enterica* isolate genomes from environmental samples in Blantyre, Malawi (2018-2020), by serovar and sequence type.**

| Serovar and Sequence Type | Number of isolates | qPCR-positive amplification for *staG* | *In silico staG* detection from genome | *In silico tviB* detection from genome |
|---|---|---|---|---|
| *S*. Orion ST 639 | 9 | 8 | 9 | 0 |
| *S*. Schwarzengrund ST 96 | 13 | 3 | 0 | 0 |
| *S*. Amager ST 293 | 14 | 2 | 0 | 0 |
| *S*. Cannstatt ST 2390 | 4 | 1 | 0 | 0 |
| *S*. *salamae* II D3:b:1,5 | 1 | 1 | 0 | 0 |
| Hissar | Choleraesuis | Paratyphi C | Typhisuis | Chiredzi | Rubislaw ST 3695 | 3 | 1 | 0 | 0 |

Seven *S.* Enteritidis ST11 genomes were detected within water samples. These were placed in a phylogenetic tree alongside contextual genomes from sub-Saharan Africa [5,68]. Genomes showed close relatedness to poultry and human disease associated *S.* Enteritidis isolates from Uganda and South Africa (S2 Fig) situated within the outlier cluster previously described (S2 Fig). Isolates of the outlier cluster have also been previously linked to multi-country outbreaks of drug-susceptible *S.* Enteritidis enterocolitis in Europe, associated with chicken eggs in Germany [69]. Therefore, the *S.* Enteritidis ST11 isolated in this study are likely to have the potential to cause diarrhoeal disease in animals and humans; however, in a setting with a high prevalence of immunosuppressive disease (i.e., HIV) the risk of iNTS disease is increased.

*S.* Isangi has been implicated in invasive diseases worldwide, including the US, South Africa, and China and detected in poultry farm environments in Nigeria and the UK [70,71]. In Malawi, cases of ESBL-neonatal sepsis caused by *S.* Isangi have occurred. Not only are the *S.* Isangi isolated from river and wastewater in Blantyre MDR, but they also have the ESBL determinants *blaCTX-M-15*, *blaOXA-1* and *blaOXA-10*. The increase in ESBL-producing invasive salmonellae is of concern due to the necessity for carbapenems for treatment of complicated iNTS disease, agents that are only intermittently available in Malawi.

Other serovars detected within this study have been reported to cause human infections and foodborne illness including *S.* Heidelberg, *S.* Infantis, *S.* Oranienburg and *S.* Hadar [72]. In 2019, *S.* Heidelberg ranked as the twelfth most common serovars to causing salmonellosis in the US and invasive disease has been reported [73]. Of concern are the 22 isolates carrying the resistance determinant *fosA7*, which confers resistance to fosfomycin. Often plasmid-mediated, it poses the risk of horizontal gene transfer [74].

We detected *staG* in *S.* Orion ST639, consistent with a previous *in silico* analysis [17]. Our findings therefore confirm that *staG* is not an appropriate marker to use to infer presence of *S.* Typhi in the environment. This is unsurprising given that *staG* detection is no longer considered adequate even for clinical samples, where typically only one serovar of *Salmonella* is present [17]. In the case of environmental surveillance, one is dealing with complex bacterial diversity in every sample, and multiple *Salmonella* serovars including *S.* Orion may be present in any given sample. This presents a considerable challenge to identifying a single target for a PCR. Isolates that were qPCR positive for *staG* or *tviB*, but whose genomes lacked these genes were likely from a mixed culture. These isolates were more thoroughly purified by multiple rounds of culture prior to DNA extraction for whole genome sequencing. Whilst *S.* Orion, thought to be a common serovar in rural locations due to its association with cattle and birds was the only *staG*-positive *tviB*-negative serovar we identified in Blantyre, there are many serovars that have this profile [17,75].

There were some limitations to our study. We used isolates from surveillance primarily designed to isolate *S.* Typhi from rivers in Blantyre, Malawi. The specific culture methodologies used were designed to favour *S.* Typhi and select against NTS serovars. Only 301/1,042 (28.89%) isolates archived underwent sequencing. Priority was given to *staG* or *tviB*-positive samples, with only one isolate sequenced per sample, consequent upon the limited budget available for sequencing. It is therefore likely the case that we have underappreciated NTS diversity. With such a low percentage of isolates being sequenced, data such as seasonal impact on various serovars and AMR profiles cannot be ascertained with any statistical relevance and the sample size to describe seasonality would need to be considered at the conception stage of future surveillance projects.

Genomic environmental surveillance has revealed that natural water is a source of diverse salmonellae in Blantyre, with many serovars present having clinical and public health importance. WGS allows the strains common in sub-Saharan Africa to be identified and provides insights into associated antibiotic resistance genes. Environmental monitoring therefore has the potential to inform and support NTS vaccine their roll out. Environmental surveillance enhanced by whole genome sequencing can offer a more comprehensive understanding of the transmission of clinically relevant strains of *Salmonella* and AMR genes, informing human, animal, and environmental public health policy.

## Supporting information

**S1 Fig. Maximum likelihood RAxML phylogenetic tree placing S. Typhimurium ST313 study isolates in the genomic context of currently recognised lineages of ST313. Red arrow = reference genome S. Typhimurium ST313 D23580.** Rooted to *S*. Typhimurium ST19 LT2. Visualised using ITOL [56]. Twenty-nine of these isolates are genomically similar to contextual ST313 Lineage 3 described in Pulford *et al*. 2021. Of the remaining isolates, 16 matched the contextual genomes for lineage 2.0, with one isolate matching lineage 2.1.
(EPS)

**S2 Fig. Maximum likelihood RAxML phylogenetic tree placing S. Enteritidis study isolates within the genomic context of previously sequenced S. Enteritidis ST11 isolated in sub-Saharan African [46,56].** This tree is rooted to *S*. Gallinarum (Accession number SAMN08796416). *S*. Enteritidis P125109 is used as a reference genome, shown by the red arrow. Epidemic clades as previously described are used to colour the tree [5]. All isolates of *S*. Enteritidis ST11 matched the contextual genomes described in Feasey *et al*., 2016 and Perez-Sepulveda *et al*., 2021. These contextual strains have close relatedness to poultry and human disease associated *S*. Enteritidis isolates from Uganda and South Africa.
(EPS)

**S1 Table. List of all isolates sequenced, including Accession numbers, Sample Type, Date of Collection, GPS of Origin, Species, Subspecies, Serovar, Sequence type and presence/absence of Resistance genes and SNPs.**
(XLSX)

**S2 Table. List of sequences that originated from the same sample as multiple isolates, including those where Sequence Type was the same, but were more than 3 SNPs different.**
(XLSX)

**S3 Table. Extended Version of Table 2, listing all isolates identified by sequencing.** Frequency and percentage of serovars and sequence types (ST) isolated from water samples from Malawi. *These serovars could not be further differentiated by the SISTR pipeline.
(XLSX)

**S4 Table. Real-Time PCR details, including probe and primer sequences, Mastermix concentration, Thermocycling conditions and gBlocks sequences used for optimisation.**
(XLSX)

**S5 Table. List of contextual Genomes used for ST313 analysis in S1 Fig.**
(XLSX)

**S6 Table. List of contextual Genomes used for ST11 analysis in S2 Fig.**
(XLSX)

**S1 File. R Script for Generating Maximum Likelihood RAxML Phylogenetic Tree.**
(RMD)

## Acknowledgments

The authors would like to thank members of the ERST Project Field Research Team for their support of this work.

## Author contributions

**Conceptualization:** Jonathan Rigby, Nicholas A. Feasey.

**Data curation:** Jonathan Rigby, Catherine N. Wilson.

**Formal analysis:** Jonathan Rigby, Catherine N. Wilson, Blanca M. Perez-Sepulveda, Jay C.D. Hinton, Mathew A. Beale.

**Funding acquisition:** Nicola Elviss, Nicholas A. Feasey.

**Investigation:** Jonathan Rigby, Allan Zuza, Yohane Diness, Charity Mkwanda, Katalina Tonthola, Oscar Kanjerwa, Chifundo Salifu, Oliver Pearse.

**Methodology:** Jonathan Rigby, Catherine N. Wilson, Oliver Pearse, Nicola Elviss, Mathew A. Beale.

**Project administration:** Jonathan Rigby.

**Software:** Catherine N. Wilson, Mathew A. Beale.

**Supervision:** Chisomo Msefula, Satheesh Nair, Nicola Elviss, Mathew A. Beale, Nicholas A. Feasey.

**Validation:** Nicholas A. Feasey.

**Visualization:** Jonathan Rigby, Catherine N. Wilson.

**Writing – original draft:** Jonathan Rigby, Catherine N. Wilson.

**Writing – review & editing:** Jonathan Rigby, Catherine N. Wilson, Chisomo Msefula, Blanca M. Perez-Sepulveda, Jay C.D. Hinton, Satheesh Nair, Nicola Elviss, Mathew A. Beale, Patrick Musicha, Nicholas A. Feasey.

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
