## [Decision Letter · Decision Letter 0]

24 Jan 2025

PNTD-D-24-01094

Diversity of Salmonella enterica isolates from urban river and sewage water in Blantyre, Malawi.

Dear Dr. Rigby Jonathan,

Thank you for submitting your manuscript to PLOS Neglected Tropical Diseases. After careful consideration, we feel that it has merit but does not fully meet PLOS Neglected Tropical Diseases' publication criteria as it currently stands. Therefore, we invite you to submit a revised version of the manuscript that addresses the points raised during the review process.

Please submit your revised manuscript within 60 days (REVISION DUE on 24 March 2025). If you will need more time than this to complete your revisions, please reply to this message or contact the journal office at plosntds@plos.org. Please include the following items when submitting your revised manuscript:

We look forward to receiving your revised manuscript.

Kind regards,

Grace Adira Murilla, PhD

Academic Editor

Ana LTO Nascimento

Section Editor

Shaden Kamhawi

co-Editor-in-Chief

Paul Brindley

co-Editor-in-Chief

**Journal Requirements:**

At this stage, the following Authors/Authors require contributions: Allan Zuza, Yohane Diness, Charity Mkwanda, Katalina Tonthola, Oscar Kanjerwa, Chifundo Salifu, Oliver Pearse, Chisomo Msefula, Blanca M. Perez-Sepulveda, Jay C.D. Hinton, Satheesh Nair, Nicola Elviss, Mathew A. Beale, and Patrick Musicha. Please ensure that the full contributions of each author are acknowledged in the "Add/Edit/Remove Authors" section of our submission form.

- TM on page: 7.

3) We note that your Data Availability Statement is currently as follows: "All available data has been uploaded". Please confirm at this time whether or not your submission contains all raw data required to replicate the results of your study. Authors must share the “minimal data set” for their submission. PLOS defines the minimal data set to consist of the data required to replicate all study findings reported in the article, as well as related metadata and methods (https://journals.plos.org/plosone/s/data-availability#loc-minimal-data-set-definition).

- The points extracted from images for analysis..

4) Please ensure that the funders and grant numbers match between the Financial Disclosure field and the Funding Information tab in your submission form. Note that the funders must be provided in the same order in both places as well. State the initials, alongside each funding source, of each author to receive each grant. For example: "This work was supported by the National Institutes of Health (####### to AM; ###### to CJ) and the National Science Foundation (###### to AM).".

**Reviewers' Comments:**

Reviewer's Responses to Questions

**Key Review Criteria Required for Acceptance?**

**Methods**

-Are the objectives of the study clearly articulated with a clear testable hypothesis stated?

-Is the study design appropriate to address the stated objectives?

-Is the population clearly described and appropriate for the hypothesis being tested?

-Is the sample size sufficient to ensure adequate power to address the hypothesis being tested?

-Were correct statistical analysis used to support conclusions?

-Are there concerns about ethical or regulatory requirements being met?

Reviewer #1: Yes, but some further details are required as per my comments below.

Reviewer #2: DNA extraction and genome sequencing techniques are standard. However, because the objectives of the paper have not been spelt out, the bioinformatics analysis of the methods is not clear what it is addressing. It is difficult to map the methodology to aim and critique it whether it is the right approach to address the objectives to give the desired result.

Reviewer #3: Overall well written with detail provided. Few sections where the text could be revised and some additional information provided to improve clarity and reproducibility. The mapping to reference genomes was difficult to follow.

- L111 how were the 6 Typhi identified in the 1048 salmonella isolates?

- L113 – reference needed for previously published PCR primers. It would also be useful to explain the significance of ttr, staG and tviB and why these are used as diagnostic markers for Salmonella

- The selection strategy for the 339 isolates for WGS should be revised to ensure clarity of numbers. All the S. Typhi. and then PCR positive for some genes of interest (n = 51?) and the n = 269 randomly selected ones. From L120-124 – correct in reading that these numbers from the total 339? Please also define what is a ‘grab sample’. Potentially a figure could be included to show the selection of isolates, potentially including a high-level map on the area? Could be done instead of Table 1 as this table doesn’t include any of the information about water source.

- L154 – what reads were used that were published elsewhere? These aren’t in the supp tables provided and no citation provided.

- L170-L171. What were the parameters for Roary? In particular the threshold for a ‘core gene’. Present in 99% of isolates? Or 95%? How many core genes were included for all the Salmonella isolates? Each of the core genes were first aligned (with mafft?) and the concatenated together?

- Core SNP alignment – what was the number of SNPs in the final core SNP alignment?

- Mapping to reference genomes.

o References. Please check the details provided. FN432031.1 links to Salmonella enterica subsp. enterica serovar Typhimurium plasmid pSLT-BT, complete sequence. FN424405 links to Salmonella enterica subsp. enterica serovar Typhimurium str. D23580 complete genome as reported in Kingsley 2009 http://www.genome.org/cgi/doi/10.1101/gr.091017.109.

o How was mapping undertaken with BWA? Was it done using the GHRU pipeline (https://www.protocols.io/view/ghru-genomic-surveillance-of-antimicrobial-resista-bp2l6b11kgqe/v3)>?

o Given the complete reference genomes are available – unclear how the long read sequences were used in the mapping? Please explain. Or is it that the complete genomes for these two isolates were available at WSI?

o Any masking of phage or recombination in the ST313 and ST11 analyses?

o What were used as outgroups for the trees and how were these treated in the analyses? Please provide details.

- How were the lineages in ST11 and ST313 identified? With contextual isolates that had previously been determined as having membership of a lineage? Please provide details of these contextual isolates and ensure the appropriate literature is cited.

- How were pairwise SNP distances calculated? Please ensure citation or acknowledgement of the tool (eg github if not published).

**Results**

-Does the analysis presented match the analysis plan?

-Are the results clearly and completely presented?

-Are the figures (Tables, Images) of sufficient quality for clarity?

Reviewer #1: Yes, but some adjustments are required as per my comments below.

Reviewer #2: Because the objectives of the study were not clearly defined, it is difficult to conclude where satisfactory results have been found. Table 2 is best suited in the appendix due to the size which spans 3 pages unless, it is edited to fit on one page.

Reviewer #3: - In the first paragraph of the results – important to highlight that looking at NTS only and that the Typhi isolates were not included. Is it correct that Typhi genomes were WGS but once confirmed as Typhi – were excluded from all downstream analyses?

- L212-213. Please revise this sentence for clarity as it initially read as if there were only three subspecies of Salmonella. The authors have this information at L219-220 – this could be moved up in this paragraph before getting into the details of serovars.

- Was there any fluctuation in number of Salmonella samples detected in water sources by season over ~18 month timespan? Or not possible to tell from these data? The authors note the ST313 isolates were detected in 13/19 months of the study at L222.

- L224 – please define MDR (3 or more drug classes)? Defined later at L256.

- For the isolates where SISTR reported a mix of serovars, was and investigation undertaken to look at the surface antigen regions in the genomes? eg n = 3 Hissar | Choleraesuis | Paratyphi C | Typhisuis | Chiredzi | Rubislaw* and ST3695. Were the genome assemblies comprised of lots of contigs so could really tease apart, or was it possible to compare the regions to reference genomes to e.g. Choleraesuis and Paratyphi C. Could these isolates have a novel serovar? Especially as these three isolates are all staG in Table 4.

- L260-264 – discussion of lineages in ST313. These lineages and nomenclature 2.0 and 2.2 need to be explained to aid interpretation of results by readers unfamiliar with these lineages.

- L281-286. Some of the information about staG should be in the Introduction (L283-284).

- Table 4 – what was the in silico analysis of staG and tviB positive genomes? Were the draft genomes screened for these two genes with blast? These details should be provided in the methods. L297 states in silico PCR was done on all assemblies? This correct? Were there any isolates that were positive for staG and tviB in silico that hadn’t been previously detected with PCR in vitro?

- Figure 1 had a lot of information in it. Would suggest breaking this into 2 figures.

o The tree with the tips coloured by subspecies, serovar and ST

o AMR – details of those with any AMR mechanism detected. From the AMR table – there are 5 serovars with AMR and 1 S. salamae isolate with any AMR. Potentially show as barplots facetted by serovar where any AMR was detected. This would help to highlight that S. Isangi had lots of AMR mechanisms detected include blaCTX-M-14

o Why was ttr data not included when staG and tviB presence / absence from PCR included? This would also be nice to see visualised.

- Supplementary Fig 1-2. The authors have integrated previously published data used to infer lineage and epidemic clade. Details of the inferred lineage / clade membership should be reported for the isolates in a supp table.

- Supplementary Table 2 referred to at L211. Unsure if this is the total genomes IDs table?

o Would be useful to include a column that highlights QC pass – assume the NA in columns from Genome.Name onwards are the ones that failed?

o Which genomes were ‘duplicate’ genomes and removed? Could these be noted in this table as well.

- AMR data (supp table 3?) only has 59 rows. From Fig 1 – appears the other genomes were susceptible. However, it is important useful to include and report these data for all isolates. Likewise, the ttr, staG and tviB not reported for all and should be included in this table (or Supp table 2).

**Conclusions**

-Are the conclusions supported by the data presented?

-Are the limitations of analysis clearly described?

-Do the authors discuss how these data can be helpful to advance our understanding of the topic under study?

-Is public health relevance addressed?

Reviewer #1: Yes.

Reviewer #2: The conclusions of the work is the supported by the data presented, however, there is need for clarity of the diversity being studied. This will help in determining if indeed the diversity being envestigated has been established

Reviewer #3: The conclusions are largely supported by the results. Additional clarification of methods and results will help. The public health relevance was clearly addressed.

- New concepts introduced in the discussion – group C and group F. Should be backgrounded in the introduction if discussing.

- Limitations of the study were stated. The impact of the sequencing strategy could be explored a little more at L355.

- The strengths of this study could also be highlighted in the limitations paragraph.

**Editorial and Data Presentation Modifications?**

Reviewer #1: Some adjustments are required as per my comments below.

Reviewer #2: I recommend: 'major revisions', before accepting the paper

Reviewer #3: Title – there are two titles on the first page (L1-L2). Think the first is the full title and the second is the short running title? If not – please address.

Abstract. Suggest some edits to make more accessible to PLoS NTD readers.

- Explaining terms in abstract. Switches from Typhi (L29) to NTS (L31) without making clear the difference. Uses iNTS (L39) without explaining what iNTS is and how it relates to Typhi and NTS. The full introduction explains it nicely

- L27-28 reads more suited to the discussion as this manuscript explored the ability for environmental samples to inform genomic surveillance

- The results part of the abstract could be revised. Unclear what the contemporaneous surveillance was – should this be mentioned in methods? And L41-L42 focus on lineages within ST313 which is S. Typhimurium and iNTS.

Introduction

- L85 – Typhi and NTS serovars linked to BSI. What about Paratyphi A Band C?

Comments of figures for suggested modifications are in the results section.

**Summary and General Comments**

Reviewer #1: Rigby and colleagues detail the genetic diversity and resistance patterns of environmental NTS isolated in Malawi between 2019-2021, providing valuable data on an understudied aspect of NTS surveillance. Moreover, the study provides further supportive evidence towards the weakness of using gene staG as an epidemiological marker for Salmonella Typhi surveillance. The data are important and novel, and the study appears to have been carried out to an appropriate standard and is mostly well written. However, there are some conflicting statements, inconsistencies, and missing details that need to be addressed. I hope my comments below are helpful to the authors in improving the manuscript.

• Please detail how sites were selected for environmental surveillance as well as the timelines for surveillance of different sites. A map may be helpful here.

• Line 39: The statement “These environmental iNTS isolates did not, however, cluster with pathovariants contemporaneously associated with bloodstream infection “ appears to contradict statements made in lines 221-236. Please consider rephrasing to clarify the intended meaning.

• Lines 53 & 57: Please capitalise and italicise Salmonella

• Line 99: A space appears to be missing between “on” and “Salmonella”

• Line 111: I think a word may be missing here

• Line 115: Please include a supplementary table of PCR primers used, or a reference to a publication that details these. I believe the primers may be those detailed in reference #16 but this is not clear at present. If they are indeed those from reference #16, it may be helpful to briefly reiterate key aspects of “pathway P” and other methods from reference #16 in the manuscript text so the reader does not have to look these up to understand the approach used

• Line 144: Please consider detailing any QC cutoffs used here

• Line 153 & 159 & 162: Please detail how the QC thresholds used were selected

• Line 157: Please provide a citation for the WSI pipeline or details of the methods/software used. Please also detail which assemblies from which tools were used in the final analysis. If only assemblies from SPAdes were used, then perhaps the remaining methods details can be removed?

• Line 164: Please include a citation for SISTR

• Line 167: I was unable to find details of the cutoffs used in Table S1 – this instead appears to be a list of distinct isolates?

• Line 172: I think this sentence may need to start with “A” and “were” should be “was”?

• Line 175: Please detail the models used for tree inference e.g. GTR gamma etc.

• Lines 177-183: It’s unclear from the current text why the “duplicate” samples were pruned from the tree – was this for ease of visualisation or it to have “representative” genomes from each sample? Please clarify the rationale in the manuscript text

• Line 188-189: It is unclear why long reads are mentioned here, presumably the analysis was carried out using short reads only? If hybrid assemblies were used then details of these need to be included at the appropriate section.

• Line 192: Please include details of how recombination filtering was conduced e.g. using Gubbins (PubMedID/PMID: 25414349) or similar here

• Line 193-4: Please detail the specific outgroups used here – I realise some of this detail is captured in supplementary figure legends but it would be helpful to mention here

• Line 198: Please include a table detailing the contextual genomes used in the study. I also think “measurement” may need to be “measurements” here

• Line 209-211: I could only identify n=15 mixed samples detailed in table S2 - perhaps an extra column could be added to highlight this information more clearly? I also got the same result looking over table 2. Moreover, please note that these mixed samples should have been excluded from phylogenomic analysis, and it would be helpful to clarify this in the manuscript text

• Figure 1: This is a nice figure but please note that many of the colours used here are very similar so it is difficult to discern some serovars and STs from others. Perhaps text labels could be used here instead or only the most common types shown with colours, with all other minor types coloured a single colour for “other”? The text is also quite blurred on my copy of this figure but this may be due to the compression used by the journal system. Please also note that it may be possible to remove the tviB coloured bar from the heatmap as the gene appears absent in all samples shown. Please also note that the scale bar label overlaps the tree – this may be readily fixed in illustrator if the file is exported from R as a pdf, but can also be fixed in ggtree code.

• Line 232: Please consider providing a brief explanation of the “outlier clade” and a citation here so that readers do not have to follow up with this in other studies.

• Line 233: Please consider providing a definition of MDR in the methods section – i.e. does this refer to co-resistance to a particular combination of drugs, or resistance to a number of different drug classes? I can see this is explained at line 255 onwards, but earlier mention at first use would help prevent confusion

• Line 313: Please include comma(s) if using “however” mid-sentence

• Line 617: Please provide the accession number of the Gallarium genome used to root the tree

• Line 329 and elsewhere: Please review the use of capitalisation of salmonellae throughout for consistency.

• Line 308-315: At a SNP distance level, how closely related were the environmental NTS to those causing human infections? Can these data answer questions related to transmission dynamics raised in the discussion?

• Supplementary AMR table: Please note that a trailing ‘Ê’ character in present in all accession numbers and some lane numbers. Also, it would be helpful to see ttr, staG, and tviB data for all genome data, not just those with AMR determinants. Perhaps these data could be added to the supplementary total genome IDs table? Regarding the AMR data shown – it would be better to code these as binary for presence/absence rather than S/R as the column headings are genes and not drugs.

• Supplementary total genome IDs table: Please also include individual sample accession numbers to make it easier for readers to link the information to the genomes in ENA/GenBank. Also, as the last four columns of this table are all NA, perhaps they are not needed?

Reviewer #2: The paper analyses samples collected from the environment, in urban river (87.9%) and sewage water (8.85%) in Blantyre Malawi. The initial surveillance was for Salmonella Typhi from 2019-2021. However further investigation for Salmonella Typhimurium was done on the samples based on which this analysis was done. A total of 1,042 samples of NTS were identified of which 339 were sequenced of which 270 genomes passed quality check. The paper demonstrated that S. Typhi ca be cultured from water river samples.

The aim of the paper was to determine the diversity of NTS isolates from urban environments. The paper fits the journal but fails short of the standard required by the journal. Objectives have not been clearly defined.

Reviewer #3: - Ethics. No statement was included. Do not think required as environmental surveillance of Salmonella from water sources.

- Data availability. The authors have made a good effort at making the data available in this study. In the results section additional details were requested. Briefly these are:

o Accessions provided in AMR table (details of 59 isolates). Total genome tables has details for 341 isolates but accessions for each not provided. There look to be thousands of isolates in PRJEB37378 (the associated bioproject).

o Please ensure each of the 339 genomes in the study have full information for AMR profiles (inferred R/ S from genotype) and presence/ absence of PCR genes (ttr, tviB and staG) in the supplementary tables.

o Details of ‘reads published elsewhere’ reported at L154 that were used for context

PLOS authors have the option to publish the peer review history of their article (what does this mean?). If published, this will include your full peer review and any attached files.

Reviewer #1: No

Reviewer #2: No

Reviewer #3: No

**Figure resubmission:**
---

## [Editor Report · Decision Letter 1]

23 Jul 2025

PNTD-D-24-01094R1

Diversity of Salmonella enterica isolates from urban river and sewage water in Blantyre, Malawi.

Dear Dr. Rigby,

Thank you for submitting your manuscript to PLOS Neglected Tropical Diseases. After careful consideration, we feel that it has merit but does not fully meet PLOS Neglected Tropical Diseases's publication criteria as it currently stands. Therefore, we invite you to submit a revised version of the manuscript that addresses the points raised during the review process.

Please submit your revised manuscript within 30 days Aug 22 2025 11:59PM. If you will need more time than this to complete your revisions, please reply to this message or contact the journal office at plosntds@plos.org. Please include the following items when submitting your revised manuscript:

We look forward to receiving your revised manuscript.

Kind regards,

Grace Adira Murilla, PhD

Academic Editor

Ana LTO Nascimento

Section Editor

Shaden Kamhawi

co-Editor-in-Chief

Paul Brindley

co-Editor-in-Chief

**Additional Editor Comments:**

This is to thank all the authors for attending to the comments from the reviewers. This Manuscript is accepted for publication with Minor Revision.

1. Please clarify whether the study was undertaken in '2019-2021' OR '2019-2022'. Refer also to the Map showing the sampling sites and the period when samples were collected (Figure 1)

2. Tables and Figures require editing to comply with guidelines

The Table titles are inconsistent: Each Table should be self-explanatory from the Label, Title, Main Body and footnotes (defining the abbreviations used).

- Table 1: PCR-based serovar prediction and number of samples selected for genome sequencing

- Table 2: Frequency and percentage of serovars and sequence types (ST) isolated from Malawian environmental samples.

- Table 3: Genomes that carried antimicrobial resistance markers (n = 103)

- Table 4: Distribution of PCR targets thought to be specifically associated with S. Typhi amongst NTS isolated from the environment in Blantyre, Malawi

I suggest that the authors include the pathogen under investigation, the country and year of isolation for ease of referencing. Please note that the samples analyzed were not Malawian environmental samples but collected from Malawi. The year is critical in case of future studies in the same area.

Definition of abbreviations (e.g. NTS) should be included in the Legend/footnotes

Figures: Titles should also be complete:

- Figure 1: "Map of Sampling Sites in Blantyre, Malawi, sampled from 2019-2022". This needs to be edited.

Please clarify - "non-typhoidal salmonella" OR "non-typhoid salmonella"

Revisit the issue raised by one of the reviewers on italicizing "salmonella" in the document.

**Reviewers' comments:**

**Figure resubmission:**
---

## [Editor Report · Decision Letter 2]

11 Aug 2025

Dear Dr. Rigby,

We are pleased to inform you that your manuscript 'Diversity of Salmonella enterica isolates from urban river and sewage water in Blantyre, Malawi.' has been provisionally accepted for publication in PLOS Neglected Tropical Diseases.

Best regards,

Grace Adira Murilla, PhD

Academic Editor

Ana LTO Nascimento

Section Editor

Shaden Kamhawi

co-Editor-in-Chief

Paul Brindley

co-Editor-in-Chief

After reviewing the written submissions, I confirm that the comments have been addressed